# *FRMD7* Gene Alterations in a Pakistani Family Associated with Congenital Idiopathic Nystagmus

**DOI:** 10.3390/genes14020346

**Published:** 2023-01-29

**Authors:** Muhammad Waqar Arshad, Muhammad Imran Shabbir, Saaim Asif, Mohsin Shahzad, Larissa Leydier, Sunil Kumar Rai

**Affiliations:** 1Department of Psychiatry, Yale School of Medicine, VA CT Healthcare Center S116A2, West Haven, CT 06516, USA; 2Department of Molecular Biology, Shaheed Zulfiqar Ali Bhutto Medical University, Islamabad 44080, Pakistan; 3Department of Biological Sciences, Faculty of Basic & Applied Sciences, International Islamic University, Sector H-10, Islamabad 44000, Pakistan; 4Department of Biosciences, COMSATS University Islamabad, Islamabad Campus, Islamabad 45550, Pakistan; 5Department of Molecular Biology, Medical University of the Americas, Charlestown KN 1102, Saint Kitts and Nevis, West Indies

**Keywords:** congenital nystagmus, *FRMD7*, novel mutation, exome sequencing, consanguinity, Pakistan

## Abstract

Congenital idiopathic nystagmus (CIN) is an oculomotor disorder characterized by repetitive and rapid involuntary movement of the eye that usually develops in the first six months after birth. Unlike other forms of nystagmus, CIN is widely associated with mutations in the *FRMD7* gene. This study involves the molecular genetic analysis of a consanguineous Pakistani family with individuals suffering from CIN to undermine any potential pathogenic mutations. Blood samples were taken from affected and normal individuals of the family. Genomic DNA was extracted using an in-organic method. Whole Exome Sequencing (WES) and analysis were performed to find any mutations in the causative gene. To validate the existence and co-segregation of the *FRMD7* gene variant found using WES, sanger sequencing was also carried out using primers that targeted all of the *FRMD7* coding exons. Additionally, the pathogenicity of the identified variant was assessed using different bioinformatic tools. The WES results identified a novel nonsense mutation in the *FRMD7* (c.443T>A; p. Leu148 *) gene in affected individuals from the Pakistani family, with CIN resulting in a premature termination codon, further resulting in the formation of a destabilized protein structure that was incomplete. Co-segregation analysis revealed that affected males are hemizygous for the mutated allele c.443T>A; p. Leu148 * and the affected mother is heterozygous. Overall, such molecular genetic studies expand our current knowledge of the mutations associated with the *FRMD7* gene in Pakistani families with CIN and significantly enhance our understanding of the molecular mechanisms involved in genetic disorders.

## 1. Introduction

Congenital nystagmus (CN) is a clinical manifestation of several general and ocular conditions. Several different nystagmus types including spontaneous, triggered, and induced have been described in the literature. Specifically, congenital idiopathic nystagmus (CIN) is a collection of irregular eye movements that occur within the first six months after birth [1]. CIN, unlike other types of nystagmus, occurs independently in the absence of any other neurological or visual defect. This has led to the belief that the condition is caused by a primary defect in the brain areas involved in ocular motor control. The effect of nystagmus on vision varies; however, because of the continuous eye movement, it can be significant. The most common type of eye movement present in nystagmus is parallel (to-and-fro); however, other forms such as rotary, vertical, pendular, and jerk have also been observed [2]. These involuntary eye movements may also influence visual acuity. The intensity and frequency of nystagmus grow over time or when the affected individual stares at an object for an extended period. Also, some illnesses increase the rate of eye movements. The affected individuals have also been observed shaking their heads; however, these symptoms are only temporary and usually fade with time and age. Patients with nystagmus may also suffer from myopia, foveal hypoplasia, and amblyopia. Additionally, patients with nystagmus may also show signs of sensory abnormalities or decreased visual acuity. Leber congenital amaurosis (*LCA1,* OMIM 204000), achromatopsia (*ACHM3*, OMIM 262300), Chediak-Higashi syndrome (*LYST*, OMIM 214500), and albinism (*OCA1A*, OMIM 203100) are some eye diseases that can be linked with CIN (www.omim.org, accessed on 24 July 2021). Also, Ocular albinism (OA) and oculocutaneous albinism (OCA) are two ocular conditions that are widely associated with nystagmus. Pigment abnormalities in the skin, eyes, and hair are present in the case of individuals with OA and OCA exhibiting the nystagmus phenotype. Several genetically-heterogenous inherited patterns of CIN such as autosomal dominant (AD) (OMIM 164100) [3], autosomal recessive (AR) (OMIM 257400) [4], and X-linked (XL) (OMIM 31700) [5] are common, with the X-linked pattern of CIN (XL-CIN) being the most common; it can be either dominant or recessive [6]. X-linked mapping has revealed three distinct X-chromosome locations (*Xp11.4–p11.3* [7], *Xp22.2* [8], and *Xq26–q27* [5]) containing two genes (*GPR143* and *FRMD7*) that are primarily responsible for this phenotype. The gene that causes CIN at the *Xp11.4-p11.3* locus has yet to be identified. Mutations in the G-protein coupled receptor 143 (*GPR143)* gene, which is positioned at *Xp22.2*, are largely associated with OA [8], with nystagmus occurring as a secondary symptom. Over 100 mutations have been reported until now in the *GPR143* gene (HGMD Professional 2021.1. )The most notable and only consistent finding of *GPR143* mutations has been CIN, which has been described in CIN families. This variant form lacks the traditional phenotypic characteristics that are widely associated with OA [9,10]. This suggests that *GPR143* mutations may have a more direct role than previously believed in the pathophysiology of CIN. However, before it can be determined that *Xp22.2* is a CIN locus, albinism must be ruled out as a contributing factor through a thorough clinical evaluation [11]. Previously, it has been established that XL-CIN, which maps to *Xq26-q27* in about 50% of families, is associated with mutations in the *FRMD7* gene [12,13]. XL-CIN associated with *FRMD7* shows a lesser prevalence as compared to CIN, which has a prevalence of 2:10,000 [14]. The FERM domain containing 7 (*FRMD7)* gene is located on *Xq26.2* and has 12 coding exons. The protein structure consists of a FERM (N-terminal, Central, and C-terminal) and a FERM-adjacent (FA) domain [15,16]. Thus far, over 116 different pathogenic variants for the *FRMD7* gene have been reported, including 70 missense/nonsense variants, 22 splicing variants, 14 small deletions, 4 small insertions, and 6 gross deletions (HGMD Professional 2021.1). Individuals depicting the nystagmus phenotype associated with mutations in the *FRMD7* gene are broadly characterized under the term *FRMD7*-related infantile nystagmus or FIN, which is characterized by the presence of horizontal, gaze-dependent congenital nystagmus. Individuals have typical eye structures and display both normal colour and binocular vision, in addition to slightly decreased visual acuity [17]. Less than 15% of the affected patients have been reported to have an abnormal head posture. Affected females have relatively better visual acuity when compared to their male counterparts [18]; however, there are no major distinctions in the frequency, amplitude, or waveform of nystagmus between either gender [14].

Over half the *FRMD7* mutations are expected to cause significant protein defects due to nonsense, frameshift, or splicing. The latter may also result in the activation of cellular mechanisms involved in mRNA repair [19]. *FRMD7* expression has been found to be temporally and spatially regulated in mouse and human brains during embryonic and fetal development [20]. Studies depicting an increased expression of the *FRMD7* gene during retinoic acid (A)-induced differentiation of mouse neuroblastoma cells suggest that *FRMD7* may play a part in neuronal processes as well [21]; however, further research is required to validate *FRMD7’s* role in development.

In this study, we present a consanguineous Pakistani family (PKNYS07) that was initially observed with signs of nystagmus. The phenotype was confirmed as CIN after genetic testing and a detailed clinical evaluation, revealing a novel nonsense mutation in the *FRMD7* gene.

## 2. Materials and Methods

### 2.1. Ethical Approval

The study adhered to the principles of the Declaration of Helsinki and was approved by the Ethical Review Board (ERB) of Shaheed Zulfiqar Ali Bhutto Medical University (SZABMU), Islamabad, Pakistan (ERB # SZABMU-775). The affected family (PKNYS07), of the Khattak ethnic group, was recruited from the Khyber Pakhtunkhwa (KPK) province of Pakistan. The family’s medical history was taken, and individuals were initially observed to have congenital nystagmus. Before blood sampling, each family member provided informed written consent. A total of 2–4 mL blood samples were extracted from each participant using clean 5-mL sterile syringes and placed in 5-mL EDTA (Ethylenediaminetetraacetic acid) tubes (Becton, Dickinson and Company, New Jersey, USA). At the research facility, samples were kept at −20 °C. An in-organic (salting out) method was used to extract genomic DNA modified from Grimberg et al. [22]. The procedure was carried out step by step at room temperature. Reagents and chemicals were kept at −4 °C and tightly sealed to avoid contamination. NanoDrop^TM^ spectrophotometer (Thermo Fisher Scientific, Dover, DE, USA) was used to determine the purity and concentration of the extracted DNA samples.

### 2.2. Genetic Studies

HumanCytoSNP-12 v2.1 Beadchip array (Illumina Inc., San Diego, CA, USA) was used to map single nucleotide polymorphisms (SNPs) across the genome of all four affected family members. The TruSight One “clinical exome” sequencing panel (Illumina Inc., San Diego, CA, USA) was used to perform Whole Exome Sequencing (WES) on a single affected individual of the family PKNYS07. Agilent Sure Select Whole Exome v6 targeting was used, as well as read alignment Burrows–Wheeler Aligner (BWA-MEM, v0.7.17), InDel realignment, base quality recalibration Genome Analysis Tool Kit (GATK) v3.7.0, SNVs/InDels (GATK/Haplotype Caller), mate-pairs fixed and duplicates removed using Picard (v2.15.0), annotation with Alamut Batch (v1.10), Copy Number Variation (CNV) detection with ExomeDepth [23], and Savvy CNV [24]. Polymorphisms in the 1000 Human Genome database (http://www.1000genomes.org/) with a Minor Allele Frequency (MAF) greater than 0.05 were excluded from consideration as candidate mutations [25]. Homozygous variants were excluded from our in-house exome data from 17 healthy distinct individuals. Non-splicing junctions containing synonymous and intronic variants were also removed [26]. 

Primers for all 12 coding exons and associated intron-exon junctions of the *FRMD7* (NM_194277) gene were designed for dideoxy sequencing using the Primer3 web tool (http://primer3.ut.ee). PCR amplicons were produced using these targeted primers. To confirm the co-segregation of the *FRMD7* variant identified by exome sequencing, a targeted sequence analysis was performed using an ABI 3730 DNA analyzer (Thermo Fisher Scientific, Dover, DE, USA). Chromas Lite (http://technelysium.com.au/wp/chromas/) and CLC sequence viewer (https://www.qiagenbioinformatics.com) software were used to visualize chromatograms and align sequence reads to the reference human genome sequence [hg19] to observe base pair changes. The *FRMD7* normal gene and protein sequence were obtained from the Ensemble genome browser (https://asia.ensembl.org/index.html). The variants were searched in HGMD2021.1, ClinVar, ExAC, and gnomAD 2.1 online genomic databases. The Human Genomic Variation Society’s nomenclature was used to name the mutations (HGVS), and the pathogenicity of *FRMD7* gene variants was determined using the ACMG guidelines.

### 2.3. Clinical Examination

For confirmation of the disease status, a detailed clinical evaluation of all affected and selected normal members of the family PKNYS07 was carried out by a local ophthalmologist. Different parameters such as visual acuity (measured in LogMAR), photophobia, head dyskinesia, colour blindness, and fundus pigmentation were evaluated. Affected individuals were also assessed for any pigment abnormalities in the skin, eyes, and hair to rule out the possibility of any underlying syndromes associated with nystagmus.

### 2.4. Bioinformatics’ Analysis

The pathogenicity of the identified variant was evaluated using various in-silico tools, such as Polymorphism phenotyping v2 (PolyPhen-2) (http://genetics.bwh.harvard.edu/pph2/), scale-invariant feature transform (SIFT) (http://sift.jcvi.org/), Mutation Taster (https://www.mutationtaster.org/), and Combined Annotation Dependent Depletion (CADD).

## 3. Results

### 3.1. Genetic Findings

Family PKNYS07 is from the Khyber Pakhtunkhwa (KPK) province of Pakistan and consists of an affected mother, unaffected father, three affected siblings (males), and four unaffected siblings (one male and three female) of different ages from the same family. The family pedigree, depicting an X-linked pattern of inheritance, is shown in Figure 1A. At the time of the initial examination, all of the affected family members had nystagmus, varying visual acuity, black hair, and normal skin colour. During the initial data collection, videos and photographs of the affected member’s eyes were amassed. Each affected person showed signs and symptoms of the disease within six months of birth. WES and genome-wide SNP genotyping were performed with the assumption that a founder variant was to be suspected; however, other genetic mechanisms were also considered. SNP genotyping using DNA from four affected family members revealed a notable (5.75 Mb from rs:5975181 to rs:6528335) shared region in the X chromosome, encompassing the *FRMD7* gene in all of the affected family members. TruSight sequencing of a single male member of the family revealed a candidate hemizygous mutation in the *FRMD7* gene, c.443T>A. The locations of the mutated, carrier, and wildtype sequences are represented by electropherograms (Figure 1B).

This variant was not present in the HGMD 2021.1, ClinVar, ExAC, or gnomAD 2.1 online genomic database. This mutation produced a truncated non-functional protein structure by introducing a premature termination codon (PTC) at position p. Leu 148 * (Figure 1C,D). For co-segregation analysis, a set of primers was designed (Appendix A), and the analysis was performed on other affected and normal family members. According to the co-segregation analysis, affected males are hemizygous, and the affected mother is heterozygous for the mutation c.443T>A; p. Leu148 *, whereas normal individuals are wild types. The co-segregation results are shown in Figure 1A.

### 3.2. Clinical Evaluation

Following genetic screening that revealed variations in the *FRMD7* gene, the affected individuals of the PKNYS07 family were revisited, and a detailed clinical examination was conducted to confirm the disease status (Table 1). Patients from the PKNYS07 family had varying visual acuities (measured in LogMAR) in both eyes and displayed the nystagmus phenotype. Among all of the affected individuals of the family PKNYS07, only one affected female (III:1) and one affected male (IV:1) had decreased visual acuity. The affected individuals (III: 1, IV:1, IV: 2, and IV: 3) had normal retinal vessels and fundus pigmentation, indicating no significant pathology. Head dyskinesia was found in all of the affected individuals except for the affected female (III:1). Furthermore, the affected individuals did not show signs of any pigment abnormalities, ruling out the possibility of any underlying condition such as OA or OCA. Also, all of the affected patients underwent colour vision and light sensitivity testing. Due to the unavailability of an Electroretinography (ERG) facility in the province, ERG could not be performed. In all of the affected members of the family, there was no other evidence of systemic or other ocular anomalies, as well as night blindness, mental retardation, or photophobia.

### 3.3. In-Silico Analysis

In-silico analysis for mutation scores using CADD, Mutation Taster, and the ACMG classification are listed in Table 2. SIFT and Polyphen2 analysis could not be performed due to the nature of the mutation (Nonsense). Mutation Taster predicted the variant to be ‘Disease causing’ with a CADD score of ‘37’. The variant was classified as pathogenic based on ACMG guidelines.

## 4. Discussion

CIN is an inherited eye disease that causes nystagmus (uncontrolled eye movement), and affected individuals commonly have visual problems. Thus far, seven loci for affected families have been proposed, three of which have been identified for XL-CIN.

A pedigree PKNYS07 is described with disease transmission from female to male and, therefore, is stated as having X-linked inheritance. No male-to-male transmission was observed (Figure 1A). Females with XL-CIN pedigrees related to the *FRMD7* locus exhibit varying degrees of penetrance with recessive or dominant inheritance patterns [11,27]. SNPs, distorted X-inactivation, and other factors involved in oculomotor development are potential reasons for the heterogeneity in penetrance. These characteristics may also be the reason that the same chromosomal location can sometimes be linked via both recessive and dominant X-linked pedigrees. Although distorted X-inactivation is one potential explanation for the varied penetrance observed in females, the proof for it is still debatable. When related to their unaffected spouses, it was found that clinically-affected females with *FRMD7* mutations were more susceptible to distorted X-inactivation [27], whereas another [13] study was unable to identify a significant difference in the pattern of X-inactivation between unaffected and affected carriers of the *FRMD7* mutations. The majority of the genes on the long arm of the X-chromosome, including those directly next to *FRMD7*, are vulnerable to X-inactivation; further, because X-inactivated genes frequently form domain-like clusters, it is very likely that *FRMD7* is also inactivated as well [11]. Generally, XL-CIN shows a dominant form of inheritance with full penetrance in males and incomplete penetrance (50%) in females. Affected males (Hemizygous) will pass on the mutated allele to daughters (Heterozygous), who may or may not display the phenotype, but not to any of their sons, whereas women with one mutated allele have a 50% chance of passing on the pathogenic variant in each pregnancy [13,14]. Carrier females generally remain unaffected; however, some deviation from this pattern has also been observed, where a sub-clinical form of nystagmus may be present but the actual cause of the disease’s partial penetrance in females is still unknown. In the case of the heterozygous female (III:1) of the family PKNYS07, incomplete penetrance was observed where the nystagmus phenotype was present due to the presence of a single mutated allele (Figure 1B), similar to the affected females (II:7) in the study depicted by Huang et al. [28]. Clinical analysis of all of the affected family members revealed the presence of the nystagmus phenotype. Iris abnormalities such as foveal hypoplasia have also been reported in patients with nystagmus [29]; however, none of the affected members of the family PKNYS07 displayed this condition. Visual acuity differed among all family members. Additionally, a horizontal waveform was present in all of the affected individuals. Also, all of the family members showed de-pigmented iridies; however, there was no underlying pathology present that could indicate the presence of a syndrome or any other disease. Thus far, only *FRMD7* has been identified as the disease’s causative gene in the affected members of family PKNYS07. The *FRMD7* gene has been found to be expressed in a variety of human tissues and to play an important role in the development of embryonic brain functions such as eye movement control [30]. Using a qPCR approach for determining the FRMD7 protein expression level in fetal tissues, Tarpey et al. [19] found significant expression in kidney and liver tissues, whereas low levels of the protein were present in brain and heart tissues. Similarly, during neuronal differentiation, FRMD7 knockdown resulted in altered neurite development, suggesting that FRMD7 plays a key role in different aspects of neuronal development [21]. In-vivo studies to determine FRMD7 interactions have revealed it to be involved in promoting mouse neuroblastoma cell growth by being involved with calcium/calmodulin-dependent serine protein kinases (CASK) [31], which is suggestive of its association with the development of the retina and neural cells [29]. Moreover, in a recent study, Jiang et al. [32] found that FRMD7 may be involved in the development of infantile nystagmus syndrome by closely interacting with GABRA2 (Type 2-γ-aminobutyric acid receptor), by using *Caenorhabditis elegans* as a model organism.

Until now, the *FRMD7* gene has never been identified as a cause of CIN in Pakistani communities. The novel nonsense variant *c.443T>A; p. Leu148 ** in the *FRMD7* gene identified by the WES approach has undermined the cause of CIN in Pakistani communities. After genetic analysis, a detailed clinical evaluation was done to confirm the disease status (Table 1). The in-silico analysis of altered amino acid sequence (Figure 1C,D) and algorithmic mutation effect prediction (Table 2) further validate the identified pathogenic variant, establishing the genotype–phenotype correlation in family PKNYS07.

Seven chromosomal loci (NYS1–7) have been linked to the development of idiopathic nystagmus (Appendix A), with only *FRMD7* and *GPR143* as the genes responsible. The Human *FRMD7* (ENSG000001656940) gene located at *Xq26.2* has 12 coding regions (ENST00000298542) and 714 amino acid residues (ENSP00000298542). There have been over 116 different pathogenic mutations reported for the *FRMD7* gene since the first reported sequence variant in 2006 [19] (Appendix A). Among them, 50% of XL-CIN pedigrees in the Western population and 47% in the Chinese population had *FRMD7* mutations [33]. Missense/nonsense mutations account for more than 60% of the reported mutations in the *FRMD7* gene, with the remaining 40% belonging to other types such as splicing and insertion/deletion. These mutations may impair FRMD7 function by disturbing the protein structure, interfering with binding sites, and preventing regulatory alteration to the proteins, such as glycosylation or phosphorylation. Tarpey et al. [19] identified that some specific mutations such as p.C271Y, p.A266P, and p.L142R are likely to affect the protein structure by introducing amino acid residues that are larger within specific areas of the protein, disrupting the helical region in the FERM domain when they were modelled on the 3D-structures of the cytoskeletal protein. Wang et al. [18] described the identification of 14 novel mutations (5 in *GPR143* and 7 in *FRMD7*) in 29 patients from eight Chinese families with CIN. The *FRMD7* mutations were localized in exons 9 and 12, whereas the *GPR143* mutations were concentrated within exons 1,2,4, and 6. Similarly in a recent study, Huang et al. [28] reported 28 novel mutations in the *FRMD7* gene from 56 pedigrees belonging to the Chinese population. Most of the mutations were present in the exons 7,8, and 9 regions encoding the FERM-C domain.

During development, the recruitment and stimulation of tiny GTPases called Rho and their regulators, GTPases activating proteins (GAPs) and Rho guanine nucleotide exchange factors (GEFs), is an important step involved in the migration of neuronal growth cones. Since the FRMD7 protein promotes neuritis lengthening and is expressed in the actin-rich distal ends of growth cones, it is thought to regulate cytoskeletal dynamics and modulate growth cone guidance primarily via Rho GTPase signaling [11,34]. Mutations in the *FRMD7* gene may inhibit the activation and recruitment of the Rho family of small GTPases along with their regulators, the Rho GAPs and GEFs [35]. Furthermore, *FRMD7* mutations in CIN may inhibit neurite process elongation during differentiation. Also, *FRMD7* mutations may impair axon direction change in response to stimuli. Thus, nystagmus may be caused by abnormal dendritogenesis and axogenesis in the regions of the brain involved in eye movement control [36].

In this study, we presented a new nonsense variant in the *FRMD7* gene, *c.* 443T>A; *p. Leu148 **. The identified nonsense mutation *c.* 443T>A; *p. Leu148 ** disrupts both the function and structure of the *FRMD7* protein by introducing a premature stop codon at position *p.148*, resulting in a truncated protein and leading to the formation of an unstable non-functional structure. Usually, in the case of eukaryotes, Nonsense Mediated Decay (NMD) is an mRNA quality control mechanism that exists to reduce gene expression errors by eliminating mRNA transcripts that contain premature stop codons caused by nonsense mutations [37,38]. mRNAs harbouring PTCs that escape NMD result in the development of one-third of all inherited human disorders. Moreover, due to the presence of several variations in the NMD mechanism, different cellular targets may be more or less sensitive [17]. 

The FERM domains [15,16] are members of the 4.1 superfamily of proteins from which they drive their name: 4.1 (four point one) and ERM (ezrin, radixin, and moesin). The structure of the FERM domain of FRMD7 consists of a FERM (N-terminal, Central, and C-terminal) lying between 2 and 282 amino acids [15], and a FERM-adjacent (FA) domain lying between 288 and 366 amino acids [16] (Figure 1C). FERM domains are composed of three-lobed “cloverleaf” structures (N-terminal, Central, and C-terminal), with each lobe depicting a tightly folded structure. Lobe A (N-terminal) has a fold similar to ubiquitin; lobe B (Central) has a fold similar to acyl-CoA-binding proteins; and lobe C (C-terminal) has a fold similar to phosphotyrosine binding (PTB) and pleckstrin homology (PH) domains [16]. The proximity of these domains suggests that they do not act independently, but rather as part of a co-ordinated structure. The ERM protein family connects the cortical actin cytoskeleton to the plasma membrane, suggesting that FRMD7 is involved in signalling between the cytoskeleton and membrane [39]. The closest homologues to FRMD7 are FARP1 and FARP2 [40], and both are involved in neuronal development, indicating a similar function for FRMD7 as well [41].

In the case of the *FRMD7* gene, most of the reported mutations occur in the regions of the FERM and FA domain, suggesting their important role in the proper functioning of the protein [42,43,44,45,46,47,48,49,50]. The FERM domain is crucial for membrane association as it binds directly to the tail of integral membrane proteins and takes part in remodelling of the actin cytoskeleton [51,52,53,54,55,56]. The FA region is found adjacent to the FERM domains and contains specific regions that act as regulatory sites for phosphorylation by different kinases [57,58,59,60,61,62,63,64]. The identified mutation in this study occurs in the FERM-C region (exon 6), which has been identified as the most conserved structure in the protein, bearing no significant homology to other proteins, and regulates their binding sites with other molecules [6,65,66,67,68,69,70]; hence, this mutation resulting in a partial non-functional protein has a pathogenic effect on all of the affected members of the PKNYS07 family. Overall, novel findings from our study regarding *FRMD7* have expanded the gene’s mutational spectrum and confirmed the genetic heterogeneity of CIN in Pakistani families. Using an in-silico approach, we also predicted the consequence of *p. Leu148 ** mutation.

## 5. Conclusions

To summarize, we identified a novel nonsense variant in the *FRMD7* gene, segregating with a rare presentation of CIN in a consanguineous family. Such molecular genetic studies not only broaden the molecular spectrum associated with *FRMD7* mutations in Pakistani families but also enhance our understanding of the molecular mechanisms involved in genetic disease.

## Figures and Tables

**Figure 1 genes-14-00346-f001:**
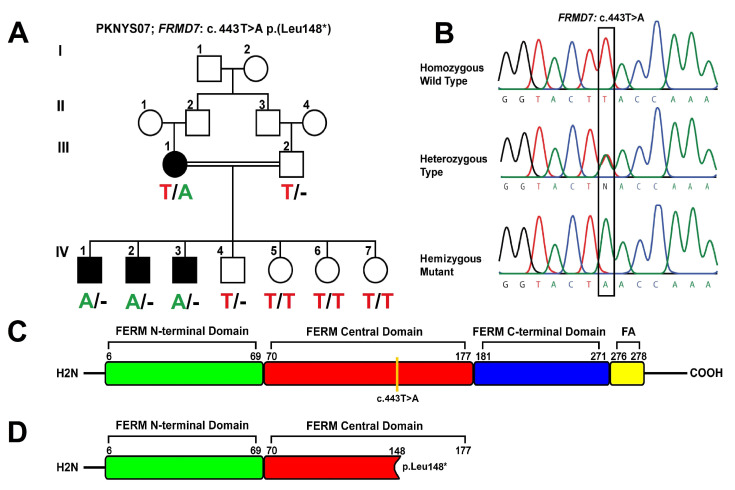
Genetic analysis of family PKNYS07 having X-linked pattern of inheritance. (**A**) Co-segregation analysis of affected, mutated, and normal individuals of PKNYS07 showing all affected males are hemizygous for c.443T>A mutation; (**B**) From top to bottom chromatograms of wildtype, heterozygous type, and hemizygous mutant individuals at position c.443; normal person presenting the base A (Top); at position c.443 heterozygous person chromatogram presenting overlapping of A/T at the base (Centre); position and mutant person’s chromatogram presenting the substitution of T>A at the same position (Bottom); (**C**) variant position concerning domain architecture of the protein. Normal *FRMD7* protein structure [15,16] consists of *FERM-N* terminal domain from 6–69 residues (green colour), *FERM*-central domain (red colour) from 70–177, *FERM-C* terminal (blue colour) from 181–271, and *FERM*-adjacent (FA) (yellow colour) from 276–278 residues. (**D**) represent mutated *FRMD7* structure showing stop codon at position *p. Leu148 **.

**Table 1 genes-14-00346-t001:** Clinical re-evaluation details of Family PKNYS07.

ID		III: 1	III: 2	IV: 1	IV: 2	IV: 3	IV: 5	IV: 6
Age (Years)		37	55	25	23	21	18	15
Gender		Female	Male	Male	Male	Male	Male	Female
Province		KPK *	KPK *	KPK *	KPK *	KPK *	KPK *	KPK *
Hair colour		Black	Black	Black	Black	Black	Black	Black
Skin colour		Normal	Normal	Normal	Normal	Normal	Normal	Normal
Visual acuity (logMAR)	Right Eye	0.6	0.0	0.70	0.0	0.0	Not available	0.0
	Left Eye	0.4	0.0	0.60	0.0	0.0		0.0
Iris colour		De-pigmented	De-pigmented	De-pigmented	De-pigmented	De-pigmented	De-pigmented	De-pigmented
Photophobia		Absent	Absent	Absent	Absent	Absent	Absent	Absent
Head nodding		Yes	No	Yes	Yes	Yes	No	No
Colour blindness		No	No	No	No	No	No	No
Nystagmus		Present	Absent	Present	Present	Present	Absent	Absent
Waveform		Horizontal	N/A	Horizontal	Horizontal	Horizontal	N/A	N/A
Convergence		No change	No change	No change	No change	No change	No change	No change
Head dyskinesia		Not found	Not found	Present	Present	Present	Not found	Not found
Age of onset		Early infancy	N/A	Early infancy	Early infancy	Early infancy	N/A	N/A
Fundus		Normal	N/A	Normal	Normal	Normal	N/A	N/A

***** Khyber-Paktunkhwa.

**Table 2 genes-14-00346-t002:** Mutation identified in this study.

Family	PKNYS07
Nucleotide Variant	c.443T>A
Protein Variant	p. Leu148 *
Status	Heterozygous (Affected Mother) Hemizygous (Affected Sons)
Type of Mutation	Nonsense
Previously reported	Novel (This study)
ACMG classification	Pathogenic (PVS1, PM2,PP1)
Mutation taster	Disease causing (1.000)
CADD score	37
SIFT	-
Polyphen-2	-

## Data Availability

All supporting data is included in the article.

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
