# Peer review of "FRMD7* Gene Alterations in a Pakistani Family Associated with Congenital Idiopathic Nystagmus"

_genes, 2023, doi:10.3390/genes14020346_

Round 1

Reviewer 1 Report

In this manuscript, Arshad et al reported a consanguineous Pakistani family affected with X-linked nystagmus, associated with a novel nonsense variant in FRMD7. The authors used bioinformatics tools and created a 3D-structure model of the wild-type and mutant FRMD7 protein. Few changes can be done to improve the manuscript:

·         In general, some sentences are misplaced in Material and Methods, Results or Discussion sections.

·         Lines 58-59: “initially diagnosed with nystagmus”. However, genetic diagnosis confirms clinical diagnosis, isn’t it?

·         Ophthalmic examination/clinical assessment is missing in Materials and Methods section.

·         Could you add a paragraph in Introduction section, explaining why the iris/hair/skin color is important to study, as you did in Results section?

·         In Materials and Methods section, the authors used the FRMD7 reference sequence: NM_194277. However, this reference encodes a 714 amino acid protein. The disease-causing variant reported in this manuscript should then be annotated: c.443T>A p.(Leu148*). With the same idea, protein scheme must be updated consequently.

·         What is minor allele frequency for this novel nonsense variant in 1000Genomes database? How many are homozygous or hemizygous for this variant in this database?

·         Visual acuity (Table 1) must be more detailed, in LogMAR.

·         All the family members shared depigmented iris. Could you discuss this, please?

·         In figure 1, what do you conclude with amino-acid conservation across species for a nonsense variant? Could you add “FA” meaning in the legend and cite where you find the position of each protein domain, please?  

·         Line 155-156: the authors wrote: “Individuals (VI/2 and IV:3) had normal retinal vessels and fundus pigmentations, indicating no significant pathology”. What about affected individuals III:1 and VI:1?

·         The authors wrote: “The Ensemble genome browser was used to retrieve 699 amino acid sequence in FASTA format” (line 171-172): why only 699 amino acid sequence was used, while the protein is a 714-amino-acid protein?

·         In the discussion, the authors discuss the consequence of this nonsense variant, “leading to an unstable non-functional structure” (lines 233-236). In the following lines (236-239), nonsense-mediated decay was mentioned. Some references must be added (PMID 26787741, 27618451). Is the premature stop codon sensitive to NMD? Could you please discuss the consequence of the novel nonsense variant identified in this family and update figures 1 accordingly?  

·         3D modelling approach is missing in Materials and Methods section. Moreover, perform a 3D model for nonsense variant is not relevant. Figure 2 should be removed.

·         Some patients with FRMD7 variants were reported with foveal hypoplasia (PMID: 24688117). Do the reported affected patients have this feature? If not, could you add this information and discuss it in the manuscript, please?

·         Seven loci were reported to be associated with CIN, do you know if there are patients from other Pakistani communities affected with CIN associated with variant in these other loci?

·         In Table 3, gene identified in NYS6 locus is GPR143, as suggested in reference 17. This Table is not essential.

·         Previous articles reported incomplete penetrance in families with FRMD7-associated CIN (PMID: 17846367, 17397053). What about your case?

·         In Table 2 and Table S2, 3-letter code should be used than 1-letter code for variant nomenclature at protein level. Moreover, Table S2 can be simplified.

Reviewer 2 Report

There is a plurality in the aim of study that leaves readers bewildered !!! In other words, it is not clear that this study presents a new variation that is expected to have a significant impact or shows the Impact of molecular-based diagnostic tests or even highlights the role of the genetic counseling process???

I highly recommend that the manuscript must be presented from a short communication aspect (a novel variation that is expected to have a significant impact on disease which has been revealed by genetic tests).

Round 2

Reviewer 2 Report

Accept in present form

Author Response

Reviewer 2 comments have already been addressed.